# MR-Guided Hypofractionated Radiotherapy: Current Emerging Data and Promising Perspectives for Localized Prostate Cancer

**DOI:** 10.3390/cancers13081791

**Published:** 2021-04-09

**Authors:** Francesco Cuccia, Stefanie Corradini, Rosario Mazzola, Luigi Spiazzi, Michele Rigo, Marco Lorenzo Bonù, Ruggero Ruggieri, Michela Buglione di Monale e Bastia, Stefano Maria Magrini, Filippo Alongi

**Affiliations:** 1Advanced Radiation Oncology Department, Sacro Cuore Don Calabria Hospital, 37024 Negrar di Valpolicella, Italy; francesco.cuccia@sacrocuore.it (F.C.); rosario.mazzola@sacrocuore.it (R.M.); michele.rigo@sacrocuore.it (M.R.); ruggero.ruggieri@sacrocuore.it (R.R.); filippo.alongi@sacrocuore.it (F.A.); 2Department of Radiation Oncology, University Hospital Munich Campus Grosshadern, 81377 Munchen, Germany; Stefanie.Corradini@med.uni-muenchen.de; 3Medical Physics Department, ASST Spedali Civili Hospital, 25123 Brescia, Italy; 4Department of Radiation Oncology, ASST Spedali Civili of Brescia, 25123 Brescia, Italy; marco.bonu@unibs.it (M.L.B.); michela.buglione@unibs.it (M.B.d.M.eB.); stefano.magrini@unibs.it (S.M.M.); 5Radiation Oncology Department, University of Brescia, 25121 Brescia, Italy

**Keywords:** mr-guided radiotherapy, prostate cancer, stereotactic body radiotherapy

## Abstract

**Simple Summary:**

The biological features of prostate cancer as a tumor with a low alpha beta ratio have led clinicians to consider the use of higher doses per fraction, thus gaining an advantage both in terms of clinical outcomes and of logistic opportunities. To date, moderate hypofractionated schedules are supported by several international clinical guidelines. The subsequent step was represented by the adoption of extreme hypofractionated schedules, for which recent literature data report non-inferiority results for the five-fractions regimens. In this scenario, the recent introduction of MR-guided daily adaptive radiotherapy is a potential paradigm shift, given the ability to increase the resolution of the pelvis anatomy and to take into account of the daily variations in shape and size of the nearby healthy structures.

**Abstract:**

In this review we summarize the currently available evidence about the role of hybrid machines for MR-guided radiotherapy for prostate stereotactic body radiotherapy. Given the novelty of this technology, to date few data are accessible, but they all report very promising results in terms of tolerability and preliminary clinical outcomes. Most of the studies highlight the favorable impact of on-board magnetic resonance imaging as a means to improve target and organs at risk identification with a consequent advantage in terms of dosimetric results, which is expected to relate to a more favorable toxicity pattern. Still, the longer treatment time per session may potentially affect the patient’s compliance to the treatment, although first quality of life assessment studies have reported substantial tolerability and no major impact on quality of life. Finally, in this review we hypothesize some future scenarios of further investigation, based on the possibility to explore the superior anatomy visualization and the role of daily adapted treatments provided by hybrid MR-Linacs.

## 1. Introduction

Prostate cancer (PC) is the most frequently diagnosed tumor in the male population in Europe [1], with high survival rates. Besides surgery, radiotherapy (RT) represents the best non-invasive alternative in the curative setting and plays a key role in the post-operative scenario [2,3,4].

There are several data supporting PC as a tumor with a low alpha-beta ratio that is more sensitive to higher doses per fraction [5,6,7].

This biological characteristic is the main basis for the worldwide propagation of hypofractionated schedules, which were initially supported by a large number of studies and are currently implemented in several international clinical practice guidelines [8,9,10,11].

The excellent outcomes in terms of toxicity and disease control and the constant technological advances have led clinicians to investigate the use of extreme hypofractionation, which combines a superior biological effect with non-negligible logistic advantages [12,13]. To date, very promising results are available in the literature and the role of extreme hypofractionation is expected to gain more attractiveness with the recent introduction of Magnetic Resonance (MR)-guided RT performed with Linacs equipped with on-board MR-imaging [14,15,16,17].

The advent of these hybrid machines may represent a game-changer for the radiation oncology community, aiming to improve the accuracy in target volume and organs at risk (OARs) delineation, based on a better anatomy visualization due to the improved soft tissue contrast provided by MR. Because the prostate can be clearly identified using MRI, it is expected that target volumes will decrease, also inter-observer variability will be reduced in accordance with ESTRO-ACROP guidelines [18,19].

Moreover, MR-Linacs allow a daily online treatment plan adaptation based on the ability to recalculate the plan prior to each fraction, taking into account changes in shape and size of the target and surrounding healthy structures [20].

These advantages could significantly reduce the inter-fraction variability, which is a major problem in extreme hypofractionated schedules [21,22].

In contrast, the longer duration of the treatment session can potentially affect intra-fraction motion, although cine-MR sequences allow clinicians to constantly monitor organ motion during the beam-on-time and apply automated beam gating features, where available [23].

However, as recently reported by Hehakaya et al. [24], the setting of PC is a congenial field for the development of MR-guided RT, given the opportunity to improve treatment tolerability with a potentially lower incidence of toxicity and a consequently favorable outcome in terms of patient-reported outcomes (PROMs). Moreover, the implementation of these hybrid devices represents a theoretical opportunity that also has positive socio-economic implications, both in terms of professional developments and for logistical reasons. Furthermore, specifically in the setting of prostate cancer, but also generally speaking, the improved accuracy in target volume delineation and the possibility to daily-adapt the target based on real-time anatomy will increase clinicians’ confidence in proposing extremely hypofractionated schedules with a reduced length of the treatment and decreased accesses to the facility. Indeed, this device is expected to reinforce the multi-disciplinary nature of RT by involving multiple professional groups, such as radiologists, physicists and Radiation Therapy Technologist (RTTs), and leading to a new dynamic in daily clinical activity [25,26,27].

Given the relative novelty of this technology, several diagnostic and therapeutic opportunities can be explored, especially in the setting of PC, such as radiomics or focal boost investigational studies in order to further tailor the oncologic treatment. Nevertheless, to date, the published evidence remains quite sparse [28,29]. In this narrative review, we aim to present the preliminary data currently available in the scientific literature on the implementation of hybrid Linacs with on-board MR-imaging for daily-adaptive stereotactic body radiotherapy (SBRT) of the prostate and the future potential challenges arising from the introduction of this technique.

## 2. Literature Research

We have performed a narrative review of the available literature concerning the of hy brid Linacs equipped with on-board MR-imaging for daily adaptive stereotactic body radiotherapy for prostate cancer. In December 2020, we started a PubMed literature search using the following research terms: “mr-guided” (MeSH terms) OR “mr guided” (All Fields) AND “daily-adaptive” (MeSH terms) OR “daily adaptive” (All Fields) AND radiotherapy” AND “stereotactic body radiotherapy” (MeSH terms) AND “prostate cancer” (MeSH terms). Only articles published in English language from peer-reviewed journals from were considered. Additional references were extracted by a hand search on the bibliography of all the selected articles.

## 3. MR-Guided Radiotherapy: Present Evidence

To date, two MR-Linac devices are commercially available, Unity Elekta (Elekta, Stockholm, Sweden) and MRIdian Viewray (Viewray Inc., Cleveland, OH, USA) [30,31].

Unity Elekta conjugates a 1.5 T magnetic resonance system with a 7 MV linear accelerator and it allows daily-adapted radiotherapy by means of two different workflows: the adapt-to-position (ATP) procedure is based on a daily update of the iso-center position, with no need for re-contouring, while in the adapt-to-shape (ATS) workflow, the daily treatment plan is re-calculated on the re-contoured volumes of the real-time anatomy of the patient (Figure 1).

The MRIdian Viewray combines a 0.35 T split magnetic resonance scanner with a circular ring-gantry in which all 6 MV Linac components are shielded to avoid magnetic field interferences. This hybrid machine enables also different types of plan adaptation ranging from simple re-optimization to a full online-adaptive workflow with re-contouring and dose re-optimization. Moreover, it allows real-time soft tissue tracking and gating.

For both devices, given the relatively longer treatment time per session, the simulation process is a crucial factor in order to perform an accurate and refined treatment delivery. Based on available literature, most experiences reported a similar protocol in terms of bladder filling and rectal emptying ([32,33,34,35,36,37,38]—see Table 1). For both the CT scan (performed for dose calculation purposes) and the MRI scan, patients were educated to have a half-full bladder in order to take into account residual volume changes during the plan adaptation phase (Figure 2).

A T2-weighted gradient-echo sequence is acquired for a better visualization of the prostate gland. After the re-optimization of the plan, a further cine MR, usually acquired on sagittal and coronal planes, is performed to check organ motion during the beam-on time (Figure 3).

The Viewray system, in addition to T2-weighted imaging, currently has a True Fast Imaging with steady-state-free precession (TRUFI) sequence. The system enables simple couch shifts, as well as more elaborated online plan adaptation strategies [40].

Currently available evidence reports MR-guided SBRT as a safe and feasible treatment option. Alongi et al. [36] reported excellent preliminary results in terms of PROMs in a cohort of 25 patients who received 35 Gy in 5 fractions, with no evidence of acute G ≥ 3 adverse events. Interestingly, the favorable results in terms of quality of life (QoL) outcomes after a median treatment time of 56 min (range, 34–86) per fraction, indicate the tolerability of MR-guided SBRT for prostate cancer and show that the longer treatment time per session has only a minimal impact on QoL. In agreement with these findings, also the study by Bruynzeel et al. [32], performed using a 0.35 T MR-Linac, reported early promising results in a phase II study enrolling 104 patients, with only 5.9% of grade 3 genitourinary toxicity according to RTOG criteria. Similarly, on QoL evaluation, no relevant differences were detected at any time point of the study, with the exception of role functioning. These data were recently updated with a final PROMs analysis after one year of follow-up, which confirmed the absence of G ≥ 3 adverse events. Furthermore, at 12 months after the end of treatment, QoL returned to baseline conditions, with only 2% of patients reporting persistent bowel symptoms [33].

A further recent paper has been published by Uguerler et al. [38] reporting in a series of 50 patients with a median follow-up of 10 months with no evidence of G3 acute or late toxicity. Although observing a 36% rate of G2 GU adverse events, when available, late GI and GU toxicity rates were respectively 2% and 6%.

In this scenario, the use of rectal spacers for mitigating prostate motion represents a helpful tool to maximize the safety and accuracy of extremely hypofractionated treatments for prostate cancer [41,42]. To date, the use of this device has been safely reported by Alongi et al. in a series of 20 patients who received MR-guided prostate SBRT using rectal hydrogel spacer. Interestingly, the authors recorded a significant advantage in terms of rectal sparing and target coverage, in comparison with a cohort of patients who did not receive the administration of the rectal spacer. In addition, despite the invasive procedure, no adverse impact on QoL was observed using PROMs assessment [43].

The same sample of patients was also analyzed in a subsequent study with the aim of evaluating a potential positive effect in terms of intra-fraction motion mitigation. The authors recorded a statistically significant effect of the rectal hydrogel spacer on rotational antero-posterior displacements compared to patients without spacers. Although these data are preliminary, they suggest a potential effect of prostate fixation due to the squeezing effect towards the pubic bone, but mature evidences is still needed to support a potential clinical impact of these dosimetric advantages [44,45,46].

Consistent with this, the study by Nicosia et al. [39] also highlights the beneficial impact of a superior anatomy visualization provided by MR-guided radiotherapy. In a dosimetric comparison between 40 patients receiving prostate SBRT using MR-Linac or a Volumetric Modulated Arc Therapy (VMAT) technique with or without fiducials, the authors recorded a significantly lower rate of constraints violation in the MR-Linac cohort compared to Volumetric Modulated Arc Therapy - Image Guided Radiation Therapy (VMAT-IGRT) patients treated without fiducials. Thus, the authors suggest that in VMAT-IGRT, only the implementation of fiducials can lead to a comparable quality in terms of real dose-distribution, which consequently highlights the advantages of MR-Linacs as a fiducial-free technique for extreme prostate hypofractionation.

## 4. MR-Guided Radiotherapy: Future Directions

### 4.1. Boost of the Dominant Intraprostatic Lesion

Despite the limited literature currently available, MR-guided SBRT in PC leads the way to several therapeutic opportunities to be explored. Among these, the administration of a boost to the dominant intraprostatic lesion (DIL), defined as the largest radiologically detected nodule in a milieu of a multifocal disease, is a critical issue for the RT scientific community [47,48]

Sparse emerging evidence suggests that the administration of doses ≥90 Gy_EQD2_ to the dominant macroscopic node has a potentially favorable impact on biochemical control and biochemical disease-free survival. Furthermore, the administration of a higher dose to the DIL is thought to improve biochemical and local control, based on evidence reporting the macroscopic dominant nodule as the first site of local relapse after curative radiotherapy [49,50,51,52].

In these series, the boost delivery was performed using a variety of techniques, including External Beam Radiotherapy (EBRT), SBRT and brachytherapy. Interestingly, only the ASCENDE-RT trial reported an increased incidence of genito-urinary effects [53].

In contrast, the recently published primary endpoint analysis of the multicenter prospective HYPO-FLAME trial reported acceptable acute GI and GU toxicity rates in a population of 100 men with intermediate and high-risk prostate cancer [54]. More mature data now provide further evidence in terms of clinical benefits, including the currently ongoing FLAME phase III trial [55].

In this scenario, the ability to rely on daily MR-guided imaging allows clinicians to improve the quality of IGRT and increase the confidence in the delivery of a focal boost based on daily re-calculation of the plan that accounts for inter-fraction variability. Of note, the correct visualization of the DIL may be difficult, for example in the case of concomitant androgen deprivation therapy, also because diagnostic MRI is still superior in terms of soft tissue contrast compared with the on-board MRI of hybrid machines [56]. Moreover, as reported by van Schie et al., the T2-signal of healthy prostate decreases during radiotherapy, making the identification of the DIL more complex [57].

### 4.2. Margin Reduction/Single Shot Treatments

The exploration of hypofractionation in recent years has led clinicians to consider the possibility of introducing single fraction regimens. As this option has been preliminary reported for SBRT of oligometastases [58], it is currently under investigation also in the setting of PC SBRT. The prospective multicenter phase I/II study ONE SHOT is investigating the feasibility and efficacy of 19 Gy single fraction SBRT with urethral sparing in patients with low and intermediate risk prostate cancer [59].

To date, only phase I results have recently been published with no acute grade ≥3 toxicity reported. The study recruitment is ongoing, and phase 2 results are eagerly awaited [60].

As recently hypothesized in a dosimetric study by Dunlop et al. [61], MR-Linacs may represent the best device for the delivery of single fraction PC SBRT. The authors investigated the technical feasibility of MR-guided prostate SBRT in 5, 2 or 1 fractions and reported no constraints violations in 30 plans. Only in 4 out of 10 plans of the 2- and 1-fraction regimens, target coverage criteria in terms of PTV D95% were not met in order to comply with Organs At Risk (OARs) constraints. On this basis, the authors planned to conduct a study to evaluate the clinical feasibility of a two fraction schedule. In this dosimetric study, an isotropic margin of 2 mm was applied to generate the PTV.

As mentioned above, it remains a matter of debate whether the use of a rectal spacer can lead to a margin reduction strategy. As recently reported by Mannerberg et al. [62], the daily volume changes of the bladder and rectum result in a large displacement of the prostate, which increases the risk of a potential target underdosing. Combined with the time-consuming procedure of daily online adaptive treatments, a margin reduction in the absence of a stable immobilization of the prostate appears to be unwise at the moment.

### 4.3. Sexual Function Preservation

Given the ability of MRI to better visualize pelvic structures, in the context of prostate SBRT, there is an increased attention being paid on preserving sexual function. The refined quality of the diagnostic process has led to an earlier detection of the disease, prompting the scientific community to reflect on the optimal balance between safety and efficacy, including the occurrence of erectile dysfunction [63]. The onset of this side effect is based on a multifactorial pathogenesis that includes both organic and psychological factors [64].

To date, the biological rationale for erectile dysfunction after RT is thought to be based on a mechanism of vascular sclerosis; however, it is unclear which healthy structure is directly involved in the development of this late sequela [65].

Moreover, the radiation-induced inflammatory response of the prostate gland may potentially contribute to facilitate this injury, along with eventually concurrent ADT or the injection of rectal immobilization devices, for which conflicting data in terms of pro-inflammatory effects have been reported [66,67].

The study by Spratt et al. [68] focused attention on sparing of the internal pudendal arteries, with encouraging results. A recent review by Ramirez-Fort et al. highlights the role of the ejaculatory ducts and the neurovascular plexus; the latter is adherent to the posterior part of the prostate gland and is therefore difficult to avoid with current image-guided radiotherapy modalities. Assuming an anatomic similarity to the brachial plexus, the authors hypothesize a similar dose constraint in conventional fractionation with a Dmax<75 Gy to 2 cc ([64,65,69]—Figure 4).

Although longer treatment sessions in this setting may theoretically result in greater organ displacement with a consequent major inflammatory exposure, MR-guided RT may help clinicians in identifying these pelvic substructures with the aim of reducing the dose exposure and consequently preserve sexual function. However, further studies are needed to confirm this approach.

### 4.4. Re-Irradiation

Another potential area of interest for MR-guided prostate SBRT is local re-irradiation after curative or post-operative RT [70].

Furthermore, in this setting, solid evidence is currently lacking and generally consists of small and mono-institutional retrospective series [71,72,73,74,75,76,77,78,79,80].

Nevertheless, preliminary data are very promising in terms of toxicity, biochemical control and ADT-free survival, prompting clinicians to consider this therapeutic alternative in a scenario in which there is a lack of consensus regarding clinical management [81].

In addition, the availability of refined imaging modalities such as PET-CT with more sensitive tracers and multiparametric-MR has increased the accuracy in identifying the site of local relapse, improving the confidence in proposing a more tailored treatment [82].

Specifically for MR imaging, it should be noted that local recurrence detection can potentially be hampered by T2-signal distortions induced by the previous RT treatment; nevertheless, the integration of diffusion-weighted imaging (DWI) and dynamic contrast-enhanced (DCE) imaging is expected to overcome these limitations [83,84].

Through an online adaptive workflow, MR-guided SBRT treatments can provide a better sparing of healthy structures, which is a crucial issue especially in the setting of re-irradiation. Compared to conventional CT-based image-guidance, the MRI-based IGRT may be the optimal choice for prostate re-irradiation SBRT.

## 5. Patient Selection

Patient selection is a crucial issue for the treatment with a hybrid MR guided RT device. Several criteria can be strongly evaluated anticipatedly for each patient to address to MR-guided RT. Considering that this innovative hybrid radiation technology is complex, it requires more resources than conventional IGRT. Moreover, all patients should be accurately screened for MR compliance. Physical limitations such as the presence of non-MR compatible pacemakers or other electronic devices in the body that may interact with magnetic fields are a reason to exclude them from this treatment, as are patients with clinical limitations such as severe claustrophobia or other severe psychological disorders. Moreover, the simulation process has to include the coil in the immobilization phase, thus representing a potential issue in the case of overweight patients who may not be compatible with the field of treatment of MR-Linacs. Given the novelty of this technology and the relatively paucity of data, to date there are no standardized limitations in terms of body mass index or weight to determine the patient fitness for this technology.

Regarding extreme hypofractionation, not all patients are ideal candidates. It is known that patients with a large prostate size and prior Transurethral Resection of the Prostate (TURP) may not be ideal candidates for SBRT [85,86,87]. Similarly, patients who have significant baseline urinary symptoms may not be ideal candidates for SBRT. These baseline assessments are crucial for both conventional Linac- and MRI Linac-based prostate SBRT [88].

The high technical complexity and enormous costs of MR-guided RT technology usually limit the availability of more than one hybrid radiation delivery unit per RT department. Therefore, backup solutions are difficult to address in the event of technical problems. Additionally, some MR-guided RT treatment planning and delivery technologies are independent and stand-alone systems. Subsequently, in case of a technical failure, any attempt to treat the patient at another RT site remains challenging.

## 6. Conclusions

MR-guided RT definitely seems to be a reliable RT advancement for the treatment of PC in a variety of situations, including prostate SBRT with a daily MR-guided adaptive workflow, dose escalation strategies with or without an intraprostatic focal boost on the DIL, and re-irradiation in cases of local recurrences after prior RT. The currently available MRgRT systems are in the early stages of their potential clinical application and in the midst of continuous improvements in this field (e.g., radiomics features, etc.). During recent years, radiomic models were studied to assess PC aggressiveness, taking into account imaging textures or features that are extracted from the labeled region of functional MRI sequences [89].

Although promising, the underlying understanding of the most informative features and predictive models remains limited. Future radiomic models could predict molecular characteristics (e.g., androgen resistance) and combined with such biological features could help clinicians to better predict PC aggressiveness [90].

A new setting that will certainly increase the clinical use of MRgRT in PC in the near future is the integration of new algorithm tools (artificial intelligence) to exploit even more specific features from multiple MRI sets of functional imaging or to provide a large amount of data for an immediate application on an adaptive flow chart or even to support the decision-making process of radiation oncologists.

## Figures and Tables

**Figure 1 cancers-13-01791-f001:**
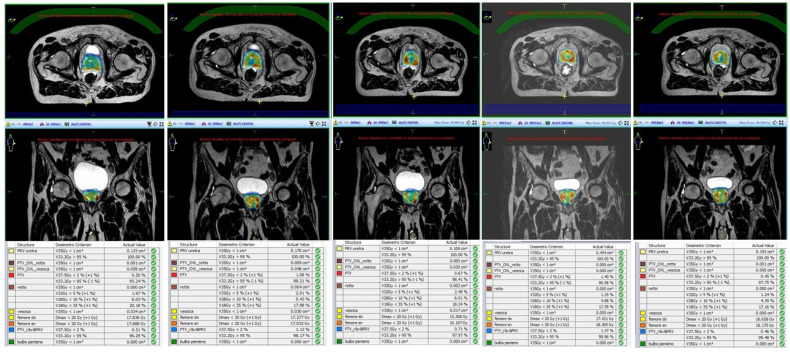
Daily replanning for Magnetic Resonance-guided Stereotactic Body Radiotherapy (MR-guided SBRT).

**Figure 2 cancers-13-01791-f002:**
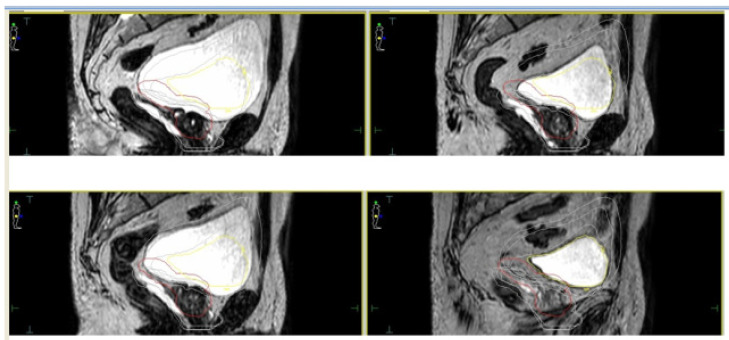
Daily interfraction variability of Planning Target Volume (PTV) and bladder.

**Figure 3 cancers-13-01791-f003:**
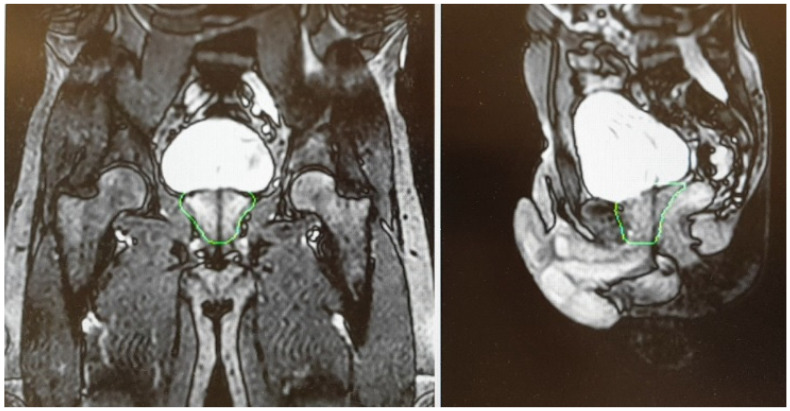
Cine Magnetic Resonance (CineMR) sequence before the delivery of MR-guided SBRT.

**Figure 4 cancers-13-01791-f004:**
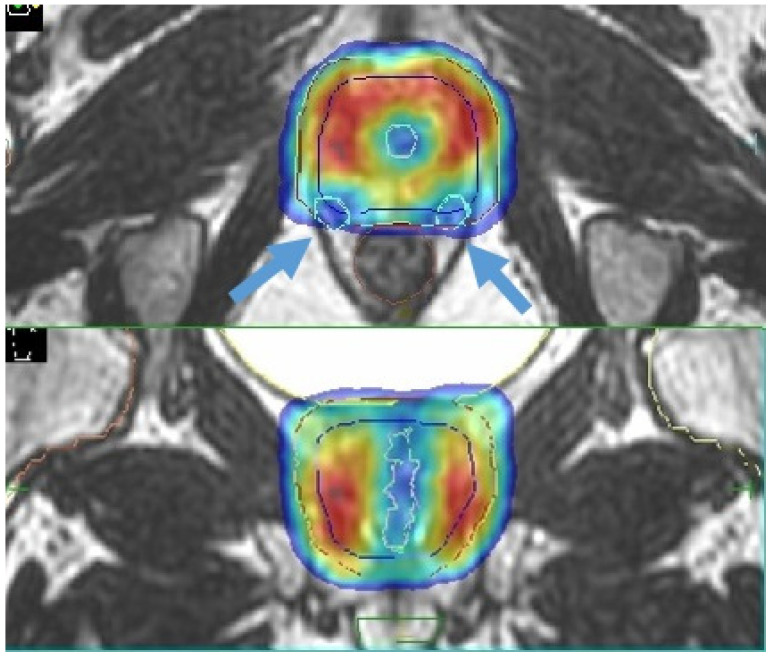
Sexual preservation during MR-guided prostate SBRT (blue arrows indicate the prostate neurovascular plexus).

**Table 1 cancers-13-01791-t001:** Literature experiences of MR-guided daily adaptive SBRT for prostate cancer.

Author	N° of Patients	MR-Linac Device	SBRT Schedule	Main Endpoint of the Study	Results
Alongi et al. [36]	20	Elekta Unity	35 Gy/5 fractions	Dosimetric analysis and preliminary PROMs report	Hydrogel improves rectal sparing with minimal impact on QoL
Bruynzeel et al. [32]	101	Viewray MRIdian	36.25 Gy/5 fractions	Early toxicity analysis	G ≥ 2 GU = 23.8% (including 5.9% of G3 according to RTOG criteria); ≥2 GI = 5.0%
Cuccia et al. [34]	20	Elekta Unity	35 Gy/5 fractions	Assessment of the impact of rectal spacer on prostate motion	Significant impact on rotational antero-posterior shifts with consequently reduced prostate motion
Tetar et al. [33]	101	Viewray MRIdian	36.25 Gy/5 fractions	PROMs analysis	After one year, only 2.2% of cases reported a relevant impact on daily activities due to GI toxicity
Nicosia et al. [39]	10	Elekta Unity	35 Gy/5 fractions	Dosimetric comparison between MR-guided SBRT and conventional Linacs SBRT	MR-guided SBRT resulted in lower constraint violation rates
Sahin et al. [37]	24	Viewray MRIdian	36.25 Gy/5 fractions	Preliminary report of feasibility	Substantial feasibility of MR-adaptive SBRT with acceptable time schedules
Ugurluer et al. [38]	50	Viewray MRIdian	36.25 Gy/5 fractions	Early toxicity analysis	Acute G2 GU = 28%; Late G2 GU = 6%; Late GI GU = 2%

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
