# Peer review of "MR-Guided Hypofractionated Radiotherapy: Current Emerging Data and Promising Perspectives for Localized Prostate Cancer"

_cancers, 2021, doi:10.3390/cancers13081791_

Round 1

Reviewer 1 Report

  1. Page 1 abstract: the authors should distinguish b/w MR-guided RT using offline MRs (MRI fused with planning CTs and delivered using standard LINACs) and combined MR-LINAC machines
  2. Page 2, para 2.  The "hybrid machines" - there are no machines referred before (unless referring to MR-guided RT being a MR-LINAC).  See comment #1
  3. Pg 2, para 6: aren't these machines a lot more expensive?  why then would there be positive socio-economic implications
  4. Pg 2, para 7: please detail your systematic review methods and findings
  5. Pg 3, para 3: spacers are more to push the rectum out of the high dose region not to reduce prostate motion (the refs provided don't talk about margin reduction)
  6. Pg 3, para 4: "prostate fixation" would require the spacer to "push" against the rectum but rectum is air filled most of the time and compliant.  Doesn't make sense from a physics point of view
  7. Pg 6, para 2: please provide refs for large prostate size and TURP being contraindications for SBRT

Reviewer 2 Report

General comments

This is well written review on a timely subject. MR-linacs are installed in the radiation oncology departments around the world. Still there is a lot of evidence lacking about the clinical benefits of the technique. This review focus on hypofractionation radiotherapy of localized prostate cancer. The current available clinical evidence is summarized. This will give a very useful scientific update as well as creating a road map for the future work to be done. The section “Present evidence” is rather short followed of “Future directions”, which has several relevant subsections of interest. There is one issue with the manuscript, which easily can be improved. It is sometimes hard to realize if the studies the authors refer to relates to MR-guided RT using MR-linac or general IGRT with/or without MR-scanner. Please be specific on this matter and state whether the studies referred to are using MR-linac.

It would be valuable if the authors can make table of the studies using MR-linac, which creates the present evidence for radiotherapy of prostate cancer.

Specific comments

page 2 “MR-guided radiotherapy”

There is a description of adapt-to-position/adapt to shape (ADP/ATS). On page 3 ADP/ATS is explained again. It would be sufficient to introduce these methods once.

page 3

The pulse sequences are not described in a similar manner for Elekta and Viewray. T2-weighted sequence is a description of the contrast not a specific sequence. However, TRUFI is a method based on the pulse sequence truefisp (or bSSFP) and not a description of the contrast. It stands for True Fast Imaging with Steady State Precession. Please, rewrite this paragraph using a correct “MRI language”.

Patient selection

This paragraph address “physical limitations”. Considering the size of the bore in MR-linacs, it would make sense to elaborate on patient weight and body mass index with respect to patient selection. Men with cancer prostate are often at an age, in which obesity is frequent. Additionally, the size of the treatment field can be quite restricted in the superior/inferior direction. This is a limitation which may affect cancer prostate treatments which include the seminal vesicles.

Reviewer 3 Report

In this submitted manuscript, the authors summarized currently available MR-guided hypofractionated radiotherapy data for treatment of localized prostate cancer. These compared MR-guided RT and SBRT with different treatment protocols. The field actually needs such retroperspective studies and perspective outlook. However I found this review is not comprehensive enough to serve as a guidance and provide sufficient insight to the field. I recommend reconsideration after the authors can address the following issues.

Reviews normally include figures and tables that clearly demonstrate the principle of these techniques, pros and cons of the technologies, different treatments etc. However this manuscript provides no one illustration for the purpose of simple readership.

Round 2

Reviewer 3 Report

The current manuscript is publishable.